# Partially Conditioned Patch Parallelism for Accelerated Diffusion Model Inference

## Abstract

Diffusion models have exhibited exciting capabilities in generating images and are also very promising for video creation. However, the inference speed of diffusion models is limited by the slow sampling process, restricting its use cases. The sequential denoising steps required for generating a single sample could take tens or hundreds of iterations and thus have become a significant bottleneck. This limitation is more salient for applications that are interactive in nature or require small latency. To address this challenge, we propose Partially Conditioned Patch Parallelism (PCPP) to accelerate the inference of high-resolution diffusion models. Using the fact that the difference between the images in adjacent diffusion steps is nearly zero, Patch Parallelism (PP) leverages multiple GPUs communicating asynchronously to compute patches of an image in multiple computing devices based on the entire image (all patches) in the previous diffusion step. PCPP develops PP to reduce computation in inference by conditioning only on parts of the neighboring patches in each diffusion step, which also decreases communication among computing devices. As a result, PCPP decreases the communication cost by around $70\%$ compared to DistriFusion (the state of the art implementation of PP) and achieves $2.36 \sim 8.02\times$ inference speed-up using $4 \sim 8$ GPUs compared to $2.32 \sim 6.71\times$ achieved by DistriFusion depending on the computing device configuration and resolution of generation at the cost of a possible decrease in image quality. PCPP demonstrates the potential to strike a favorable trade-off, enabling high-quality image generation with substantially reduced latency.

## 1 Introduction

Diffusion models (Ho et al., 2020; Song et al., 2020; Dhariwal & Nichol, 2021) are very successful in modeling and generating unstructured data, such as images, molecular structures (Huang et al., 2022), and three-dimensional (3D) models (Qian et al., 2023). In recent years, with the release of the latent diffusion model (Rombach et al., 2021), its variants (Ho, 2022; Saharia et al., 2022) have become the center of attention in both academia and industry. Doing a diffusion process in the latent space, in which dimensionality can be much smaller than the image space, requires significantly lesser computation to train a diffusion model and make inferences with it than in the image space. Despite the wide adoption and application of diffusion models, its sequential denoising process of hundreds or thousands of steps bottlenecks the inference speed and introduces non-trivial latency for interactive use cases. This challenge becomes more prevalent when the resolution of the generated images increases.

Two types of methods have been proposed to accelerate the inference of diffusion models: decreasing the number of steps required for the denoising diffusion process and increasing the computing speed of diffusion steps. The most adapted diffusion models - Denoising Diffusion Probabilistic Models (DDPMs) (Ho et al., 2020) can take thousands of denoising steps to generate an image. Various works were introduced to decrease the denoising steps required but often at the cost of the quality of generation, such as Denoising Diffusion Implicit Models (DDIMs) (Song et al., 2020) and DPM-Solvers (Lu et al., 2022a;b). Other works were introduced to decrease the computation needed for each diffusion step, such as Spatially Sparse Inference (SSI) (Li et al., 2022) and Q-Diffusion (Li et al., 2023), also trading quality for speed. With the growing interest in fast inference for diffusion models, in addition to accelerating the diffusion process on a single device, researchers began to study making inferences parallelly to generate a single image utilizing multiple devices.

ParaDiGMS (Shih et al., 2023) computes multiple diffusion steps in parallel while Patch Parallelism (PP) (Li et al., 2024) separates an image into patches and does diffusion on each patch in a dedicated device. Li et al. (2024) introduced DistriFusion, the current state-of-the-art diffusion model inference acceleration method leveraging multiple devices, which implements PP using asynchronous communication among devices.

In this paper, we empirically show that the denoising step of every patch of an image does not necessarily need to be conditioned on the entire feature map in the previous step. This allows the communication in DistriFusion to be optimized and further decreases the computation required for every diffusion step during inference. Our method **Partially Conditioned Patch Parallelsim (PCPP)** thus reduces latency compared to PP by reducing the amount of computation performed in the denoising step. As a result, PCPP decreases the communication cost by around $70\%$ compared to DistriFusion and achieves $2.36 \sim 8.02\times$ inference speed-up using $4 \sim 8$ GPUs compared to $2.32 \sim 6.71\times$ achieved by DistriFusion depending on the computing device configuration and resolution of generation.

## 2 BACKGROUND

### 2.1 DIFFUSION MODEL

Diffusion models, or diffusion probabilistic models (DPMs), are latent variable models defined by Markov chains. This stochastic model predicts the probability of a sequence of possible events occurring based on the previous event (Ho et al., 2020). The Markov chain is first trained through a forward diffusion process, during which noise is added to the sample using variational inference to control the strength of the Gaussian noise (Song et al., 2020). Subsequently, the forward diffusion process is reversed, denoising the sample at each timestep until the final image is generated (Li et al., 2023). This denoising step can be achieved through either a noise prediction model, which attempts to predict the next noise components, or data prediction models, which predict the original data based on the noise at each timestep (Lu et al., 2022b). Currently, noise prediction models that utilize the U-Net (Ronneberger et al., 2015) architecture are typically used in practice for diffusion models with attention modules (Vaswani et al., 2017) in transformers (Li et al., 2023). Let the noise prediction model be $\epsilon_\theta$. The process begins with a Gaussian noise sample $x_T \sim \mathcal{N}(0, I)$ and iteratively denoises it for $T$ steps to obtain the final image $x_0$. At each step $t$, the noisy image $x_t$ is fed into the model $\epsilon_\theta$ along with the timestep $t$ and an optional condition $c$ (such as text), to predict the noise component $\epsilon_t = \epsilon_\theta(x_t, t, c)$. The image $x_{t-1}$ at the next timestep is then calculated using $\epsilon_t$. The latency bottleneck often happens during the forward passes through the U-Net.

For conditional sampling with diffusion models, guided sampling is a popular technique for saving sampling time by reducing the retraining steps of the network. It supports two guided sampling methods for the conditional noise prediction model $\hat{\epsilon}_\theta(x_t, t, c)$. The first method is classifier guidance, which uses a pretrained classifier to define the conditional noise prediction model on the variable guidance scale. The second method is classifier-free guidance, where the unconditional and conditional noise prediction models share the same model, combining the score estimates instead of relying on a specified classifier (Lu et al., 2022b). Our focus lies on the generative denoising process that utilizes the U-Net structured noise prediction models with classifier-free guidance.

### 2.2 PATCH PARALLELISM

One approach to designing parallelism for diffusion models is Patch Parallelism (PP), where a single image is divided into patches and distributed across multiple GPUs for individual and parallel computations. In the naive approach to PP, i.e. with no communication between the patches, each device essentially creates its own full image. When brought together, these individual images form the final composite image (Li et al., 2024). To create one cohesive image using PP, synchronous communication can be employed to obtain intermediate activations between patches at each step. This approach is similar to domain parallelism, where devices communicate by exchanging halo regions at their boundaries during forward and backward convolution passes, synchronizing between communication and computation phases. However, when dealing with smaller tensors, this approach may incur significant communication overhead, which can mitigate the speedups gained from parallelizing the process (Jin et al., 2018). To address this issue for single image generation across multiple devices,

Prompt: A medical doctor in the universe, cold color palette, muted colors, detailed, 8k

| No Context Shared Partial= 0.0 | Margin Context Shared Partial= 0.1 | Minor Context Shared Partial= 0.3 | Half Context Shared Partial= 0.5 | Major Context Shared Partial= 0.8 | All Context Shared Partial= 1.0 |

**Partially Conditioned Attention**

| Upper Patch | Upper Patch | Upper Patch | Upper Patch | Upper Patch | Upper Patch |
| Neighbour Patch | Neighbour Patch | Neighbour Patch | Neighbour Patch | Neighbour Patch | Neighbour Patch |

| Neighbour Patch | Neighbour Patch | Neighbour Patch | Neighbour Patch | Neighbour Patch | Neighbour Patch |
| Lower Patch | Lower Patch | Lower Patch | Lower Patch | Lower Patch | Lower Patch |

Figure 1: Example of images generated using Partially Conditioned Patch Parallelism (PCPP) with varying partial value. The images are generated with 4 GPUs using PCPP. Due to the use of classifier-free guidance, 2 GPUs are used in the denoising process to generate pixels directly. The image is separated into upper and lower patches computed by dedicated GPUs parallelly. To provide a clear comparison, warm-up synchronization steps are not applied, i.e., the computation of patches is separated at the very first step. The blue/green region refers to the activation used to compute the upper/lower region. The activation can consist of fresh activation of the local patch and partial stale activation from the last step of the neighbor patch, depending on the partial value used.

communication needs to be introduced between patches while ensuring that no extra synchronization costs are incurred in the overall processing time. This concept is explored in DistriFusion, which introduces a hidden communication element to PP (Li et al., 2024).

DistriFusion uses one synchronous communication to facilitate patch interaction, followed by asynchronous communication at each subsequent denoising step, where the slightly outdated activations from the previous step are reused to hide the communication overhead within the computation. Additionally, the process and image quality is optimized using classifier-free guidance, dividing the devices into two batches: one for performing additional computations instead of creating an extra classifier, and the other for actual image computation. Due to batch splitting, the number of devices available for each image generation is limited, as only half of the devices are utilized for generating pixels (Li et al., 2024). As such, experiments conducted on 8 GPUs would equate to distributing the image patches across 4 GPUs. To extend DistriFusion, we delve deeper into the modules involved in each denoising step. We identify and experiment with areas that can be further parallelized through the use of partial activations and optimized communication, leading to the development of PCPP.

## 3 METHOD

In this section, we introduce our PCPP for parallelizing the inference of diffusion models using multiple devices asynchronously by point-to-point communication. The key idea is to partition the image horizontally into $n$ non-overlapping patches and process each patch conditioned on only itself and the parts of its neighboring patches (from the previous step) on separate devices. This approach is based on the hypothesis that generating image patches does not always necessitate dependency on all other patches; instead, satisfactory results can be achieved by relying solely on neighboring patches. An illustration of image generation using PCPP is presented in Figure 1. To implement PCPP, we redesign the asynchronous communication in DistriFusion and adjust the input for every self-attention module in the U-Net (Ronneberger et al., 2015) to reduce computation. Section 3.2

explains how collective communication is optimized to point-to-point communication with neighboring patches, while Section 3.3 illustrates the adjustments made to the self-attention mechanism.

## 3.1 FORMULATION

Let $x_t \in \mathbb{R}^{H \times W \times C}$ denote the image at diffusion timestep $t$, where $H$, $W$, and $C$ are the height, width, and number of channels, respectively. We divide $x_t$ horizontally into $n$ non-overlapping patches $\{x_t^{(i)}\}_{i=1}^n$, with each patch $x_t^{(i)} \in \mathbb{R}^{h \times W \times C}$, where $h = H/n$. Let $\epsilon_\theta(\cdot)$ represent the noise prediction model parameterized by $\theta$, and $c$ be the conditioning information (e.g., a text prompt). We use $s \geq 1$ as the guidance scale for classifier-free guidance. We introduce a partial value $p \in [0, 1]$ representing the percentage of the neighboring patch used in the context.

**Patch update with parts of a stale neighboring context.** At each timestep $t$, we update each patch $x_t^{(i)}$ by incorporating its own state and a partial portion of the neighboring patches from the previous timestep $x_{t+1}$. Specifically, for patch $(i)$, we define its neighboring context $\mathcal{N}_t^{(i)}(p)$ as:

$$\mathcal{N}_t^{(i)}(p) = \begin{cases} \left\{ \text{upper } ph \text{ of } x_{t+1}^{(i+1)} \right\}, & \text{if } i = 1 \\ \left\{ \text{lower } ph \text{ of } x_{t+1}^{(i-1)}, \text{ upper } ph \text{ of } x_{t+1}^{(i+1)} \right\}, & \text{if } 1 < i < n \\ \left\{ \text{lower } ph \text{ of } x_{t+1}^{(i-1)} \right\}, & \text{if } i = n \end{cases} \quad (1)$$

Here, upper $ph$ of $x_{t+1}^{(i+1)}$ denotes the top $ph$ pixels (height) of the neighboring patch $x_{t+1}^{(i+1)}$, and lower $ph$ of $x_{t+1}^{(i-1)}$ denotes the bottom $ph$ pixels of the neighboring patch $x_{t+1}^{(i-1)}$.

**Adjusted noise prediction.** For each patch $x_t^{(i)}$, we compute the adjusted noise prediction $\hat{\epsilon}_\theta(x_t^{(i)}, \mathcal{N}_t^{(i)}(p), t, c)$ using classifier-free guidance:

$$\hat{\epsilon}_\theta(x_t^{(i)}, \mathcal{N}_t^{(i)}(p), t, c) = \epsilon_\theta(x_t^{(i)}, \mathcal{N}_t^{(i)}(p), t) + s \left( \epsilon_\theta(x_t^{(i)}, \mathcal{N}_t^{(i)}(p), t, c) - \epsilon_\theta(x_t^{(i)}, \mathcal{N}_t^{(i)}(p), t) \right) \quad (2)$$

where $\epsilon_\theta(x_t^{(i)}, \mathcal{N}_t^{(i)}(p), t)$ is the unconditional noise prediction, incorporating the partial neighboring context, and $\epsilon_\theta(x_t^{(i)}, \mathcal{N}_t^{(i)}(p), t, c)$ is the conditional noise prediction.

**Reverse diffusion step.** Using the adjusted noise prediction, we compute the mean $\mu_\theta(x_t^{(i)}, \mathcal{N}_t^{(i)}(p), t)$ for the reverse diffusion process:

$$\mu_\theta(x_t^{(i)}, \mathcal{N}_t^{(i)}(p), t) = \frac{1}{\sqrt{\alpha_t}} \left( x_t^{(i)} - \frac{\beta_t}{\sqrt{1 - \bar{\alpha}_t}} \hat{\epsilon}_\theta(x_t^{(i)}, \mathcal{N}_t^{(i)}(p), t, c) \right) \quad (3)$$

where $\alpha_t$ and $\beta_t$ are the noise schedule parameters, and $\bar{\alpha}_t = \prod_{s=1}^t \alpha_s$.

The updated patch $x_{t-1}^{(i)}$ is then obtained by:

$$x_{t-1}^{(i)} = \mu_\theta(x_t^{(i)}, \mathcal{N}_t^{(i)}(p), t) + \sigma_t z, \quad z \sim \mathcal{N}(0, \mathbf{I}) \quad (4)$$

## 3.2 COMMUNICATION WITH ONLY NEIGHBORING PATCHES

In DistriFusion, each denoising step $t$ includes forward passes through $L$ layers in the U-Net. At each step, it first splits the input $x_t$ into $n$ patches $x_t^{(1)}, \ldots, x_t^{(n)}$. For each layer $l$ and device $i$, when the input activation patches $A_{l,t}^{(i)}$ are received, two processes execute asynchronously: (1) On device $i$, the activation patches $A_{l,t}^{(i)}$ are reintegrated into the previous step's stale activations $A_{l,t+1}$, and then processed by the sparse operator $F_l$ (which may be linear, convolutional, or an attention layer) to carry out the necessary computations, and (2) Simultaneously, an *AllGather* operation collects

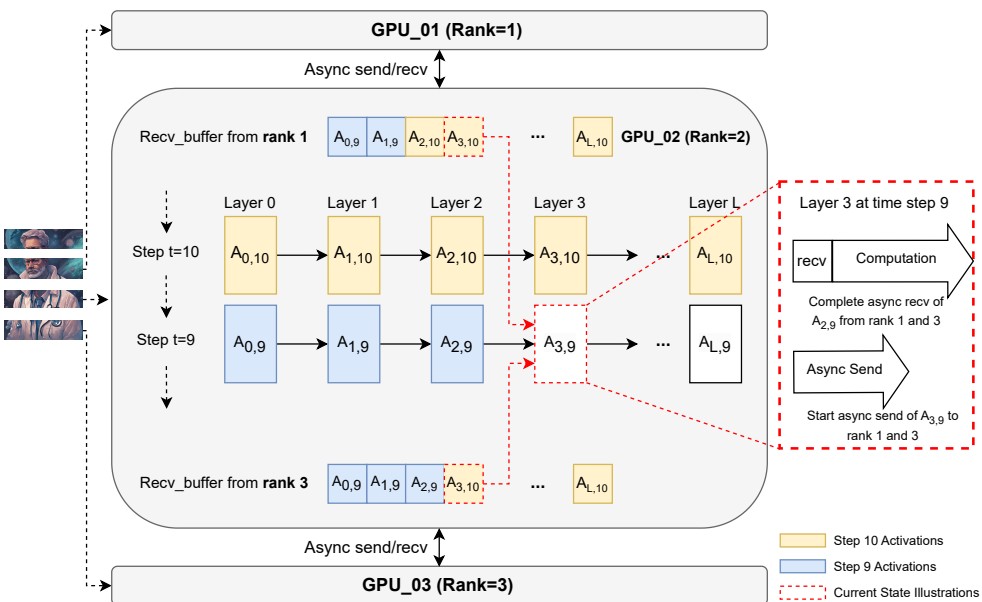

Figure 2: Partially Conditioned Patch Parallelism (PCPP) asynchronous communication design. This diagram captures the computation and *recv_buffer* state at $t = 9$ and layer 3 for device rank 2 prior to executing the self-attention computations. The device is confirming the completion of *async recv* of the previous layer's activation from its adjacent rank; once this is completed, $A_{2,10}$ in the *recv_buffer* from rank 1 will be overwritten by $A_{2,9}$ for subsequent use at time step $t = 8$. In the meantime, it initiates an (*async send*) of its own activation $A_{3,9}$ to its neighboring rank, concurrent with the self-attention computation. As described in Section 3.3, the input to the self-attention computation is $A_{2,9}^{(2)}$ and the two neighboring stale activations from last step $A_{3,10}^{(1)}$ and $A_{3,10}^{(3)}$.

$A_{l,t}^{(i)}$ from all devices to assemble the complete activation $A_{l,t}$ required for the subsequent denoising step $t - 1$.

In PCPP, we replaced *AllGather* with asynchronous point-to-point *Send* and *Receive*. Figure 2 provides a visualization of our proposed communication design during the diffusion inference phase. Specifically, each device maintains up to two receive buffers (*recv_buffer*) for activations from its two neighboring ranks, each of size $L$. After the initial warm-up steps where we perform synchronous *AllGather*, the two *recv_buffer* will be constantly overwritten by the newly received activations from its neighboring ranks. For each layer $l$, device $i$ and time step $t$, when the input activation patches $A_{l,t}^{(i)}$ are received, we perform the following steps. Firstly, wait for the asynchronous send and receive of $A_{l-1,t}^{(i)}$ to complete so that they will be saved into *recv_buffer* (while they are not used in this step, they will be used in the next time step $t - 1$). Secondly, initiate an asynchronous *send* of its incoming activation $A_{l,t}^{(i)}$ to its neighbors $i - 1$ and $i + 1$. Lastly, compute attention as described in Section 3.3 using $A_{l,t}^{(i)}$ and the two neighboring stale activations from last step $A_{l,t+1}^{(i-1)}$ and $A_{l,t+1}^{(i+1)}$, which can be directly loaded from the two *recv_buffer*. The communication overhead of the second step is hidden under the computation latency of the last step, as shown in the close-up illustration in Figure 2. The total communication cost is also greatly reduced due to the replacement of collective communication with point-to-point communication.

## 3.3 PARTIALLY CONDITIONED ATTENTION

In DistriFusion, the input to attention layers are the complete activations of the entire image assembled using stale activations from all other devices and the on-device activation as output from the previous module. In detail, the input hidden state is multiplied with matrices $W^q, W^k, W^v$ to create query, key, and value tensors. Then, the key and value tensors are expanded to include stale keys and

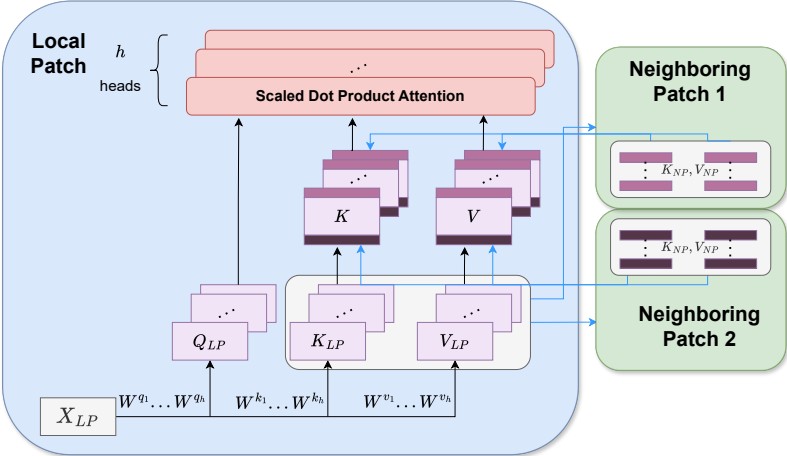

Figure 3: Overview of partially conditioned attention. The local patch receives partial $Q_{NP}$s and $K_{NP}$s from the neighboring patches. The size of partial keys and values is determined by the parameter $partial$, whose effect is shown in Figure 1. $K$ and $V$ are then assembled using $Q_{LP}$, $Q_{NP}$s and $K_{LP}$, $K_{NP}$s respectively, instead of using all the patches from the entire image. LP means local patch, NP means neighboring patch. Black arrows refer to data transfer on the local device, and blue arrows refer to communication between patches on different devices.

values from the last step from other devices by *AllGather*. We hypothesize that to generate patches that are cohesive in parallel, only neighboring context (pixels) is needed instead of the entire feature map. We test to use only a portion of the neighboring patches to generate patches in denoising steps as defined in equation (1). With this hypothesis, we propose partially conditioned attention to computing muti-head scaled dot product attention based on keys and values that consist of local context and partial neighboring context as shown in Figure 3.

Figure 1 shows how the quality of the generated image changes with different *partial* values $p$, i.e., the fraction of neighboring context used in the computation of key and value. In the case of using two GPUs to generate an image (and another two GPUs for classifier-free guidance (Ho, 2022)), if *partial* is set to 0, i.e., no neighboring context is considered when generating upper patch and lower patch in parallel, the resulting image contains two sub-images with a clear distinction between them as shown in the left-most image in Figure 1. Therefore, the computation of attention in a patch needs to be conditioned on at least a fraction of its neighboring patches to ensure a smooth connection between patches. From Figure 1, we notice that $partial \geq 0.3$ guarantees to generate a cohesive image when 2 GPUs are used. The tuning of $partial$ for different numbers of GPUs used is further discussed in later sections.

Out of the three types of layers in the U-Net—Attention, Convolution, and Group Norm—we decided to enable PCPP for self-attention only, since it involves the largest amount of data to be sent and received from other devices, as shown later in Table 2. For Group Norm, Li et al. (2024) showed that synchronization is critical for group norms to be statically useful for good generation quality. This is especially true when a large number of devices are used as group norms computed solely on on-device activations, which are insufficient for precise approximation. Hence, we used the same approach in DistriFusion instead of point-to-point communication with only the neighboring devices for computing group norms. Lastly, since the convolution layer only accounted for a small percentage of the overall communication cost and only share the boundary pixels with other devices, we decided to leave the *AllGather* function for it as it is.

## 4 EXPERIMENTS

We used the NCCL (NVIDIA Collective Communications Library) as the backend for the `torch.distributed` library to implement PCPP. In particular, we used `batch_isend_irecv` for point-to-point communication among neighboring patches. Compared with individual `isend`

Table 1: Communication cost breakdown for three different resolutions on 8 A100s. The total buffer size is the sum of Group Norm, Conv2d, and Attn components, representing the number of bytes to send to other devices during one forward pass of the entire U-Net. Compared to the method used in DistriFusion, our method PCPP reduces the communication cost by around 70%.

| Method | $1024 \times 1024$ | $2048 \times 2048$ | $3840 \times 3840$ |
|---|---|---|---|
| Total Buffer Size | 0.218G | 0.855G | 2.98G |
|     Group Norm | 0.006M | 0.006M | 0.006M |
|     Conv2d | 0.008G | 0.016G | 0.031G |
|     Attn | 0.21G | 0.839G | 2.949G |
| Total Communication Amount | | | |
|     DistriFusion | 1.526G | 5.985G | 20.86G |
|     **Ours** | **0.476G** | **1.79G** | **6.115G** |

and `irecv`, this helps avoid potential deadlocks and simplifies the bookkeeping and management of the asynchronous requests in our code. A detailed description of our parallel implementation is included in Appendix A. For the parallel inference, we used Stable Diffusion XL (Podell et al., 2024) with a 50-step DDIM sampler, classifier-free guidance applied with guidance scale $s$ of 5, and CUDA Graph to reduce launch overhead. All experiments were conducted on 2 compute nodes, each containing 4 NVIDIA A100 40GB GPUs, except for generating images of resolution $3840 \times 3840$ on 1 NVIDIA A100 80GB GPU.

To evaluate PCPP, we use the 2014 HuggingFace version of the Microsoft COCO (Common Objects in Context dataset). The COCO caption dataset consists of human-generated captions for the dataset images (Chen et al., 2015). For our experiments, we utilize a portion of the HuggingFace COCO data, from which we sample a subset of 1K images with the caption for each from the validation set. We compare the communication cost, image quality, and generation latency of our PCPP implementation to the results obtained using DistriFusion.

**Communication cost.** In PCPP, nearly all communication happens in the self-attention, group-norm, and convolution layers of the U-Net. To measure the total communication amount, we use the total number of bytes sent and received by all the devices. First, we calculated the total amount of bytes needed to send to other devices for all three layers. For the *AllGather* operation in Patch Parallelism (PP), assuming the use of ring *AllGather*, the amount is calculated as $b_s \times (n - 1) \times 2$, where $b_s$ is the send buffer size, $n$ is the number of devices, and 2 is the 2 bytes for FP16 precision. For the `isend_irecv` operations in PCPP, the communication amount is $b_s \times 2 \times 2$, where each device sends to at most 2 neighbors. Note that since we have only implemented the PCPP for the self-attention layer, we would keep the calculation for the other layers the same as in the Patch Parallelism approach.

**Image quality.** Following previous works (Li et al., 2020; 2024), we employ standard evaluation metrics, namely Peak Signal-to-Noise Ratio (PSNR), Learned Perceptual Image Patch Similarity (LPIPS), and Fréchet Inception Distance (FID). PSNR quantifies the difference between the numerical image representations of the benchmarked method outputs and the original diffusion model outputs, where higher values indicate better quality matching the original image. LPIPS assesses perceptual similarity between the two image outputs, with lower scores indicating greater perceptual similarity (Zhang et al., 2018). Finally, the FID score measures distributional differences between the two outputs, particularly evaluating the diversity and realism of the images, where lower scores indicate better results (Heusel et al., 2017).

**Latency.** The latency is calculated as the average of 20 inferences on the same prompts. Before the measurement, there are 3 iterations of warm-up on the whole denoising process. The measurements are obtained with CUDAGraph enabled to optimize some kernel launching overhead.

Table 2: Quantitative evaluation of image quality by 50-step DDIM sampler (Song et al., 2020). Four warm-up steps are used at the beginning, i.e., synchronizing the compute on the first four diffusion steps. *w/ G.T.* indicates that the metrics are calculated using the ground-truth images from the COCO dataset, while *w/ Orig.* indicates calculations using the original model's samples on a single device. For PSNR, the *w/ Orig.* setting is reported.

| #Devices | Method | PSNR (↑) | LPIPS (↓) | | FID (↓) | |
| --- | --- | --- | --- | --- | --- | --- |
| | | | w/ Orig. | w/ G.T. | w/ Orig. | w/ G.T. |
| 1 | Original | – | – | 0.796 | – | 65.3 |
| 4 | DistriFusion | **31.9** | **0.146** | **0.797** | **20.8** | 65.5 |
| 4 | **Ours** | 29.2 | 0.352 | 0.800 | 38.4 | **65.2** |
| 8 | DistriFusion | **31.1** | **0.181** | **0.798** | **24.1** | 65.6 |
| 8 | **Ours** | 28.8 | 0.403 | 0.805 | 42.2 | **64.8** |

## 5 RESULTS

### 5.1 LATENCY

Since the communication is hidden by the computation in every module, as shown in Figure 2, the latency is determined by the computation speed. By using different partial values for partially conditioned attention, we reduce the input size to the scaled dot product attention and thus speed up the computation. This effect is more salient when the resolution is higher, as shown in Figure 4. When the resolution is low, the decrease in computation is not obvious. However, when the resolution is high, the decrease in latency is more promising. Note that different *partial* for partially conditioned attention is used for 4 devices and 8 devices to ensure the quality of the image generated. When more devices are used, the height of every patch decreases, resulting in less context contained in a patch. We hypothesize that for a large number of devices, more than

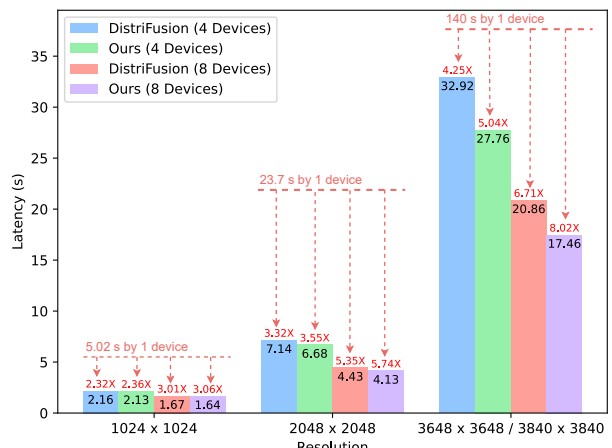

Figure 4: Inference latency for generating one image with 50-step DDIM sampler (Song et al., 2020) by resolution and device configuration. Our method sets the *partial* value for partially conditioned attention to 0.3 for 4 devices and 0.8 for 8 devices. The red numbers show the speed-up achieved compared to inference on 1 device.

the immediately neighboring patches are required for the partially conditioned attention to consider enough context.

### 5.2 COMMUNICATION COST

The results for communication cost are presented in Table 1. We followed the approach described in Section 4, which is not identical to the calculation in DistriFusion (Li et al., 2024). For consistency, we recalculated the total communication amount for DistriFusion. As expected, the point-to-point communication overhead of PCPP is significantly reduced compared to the collective communication in PP, achieving an average of about 70% reduction in communication amount. The majority of the savings come from the self-attention layer, which dominates the communication overhead.

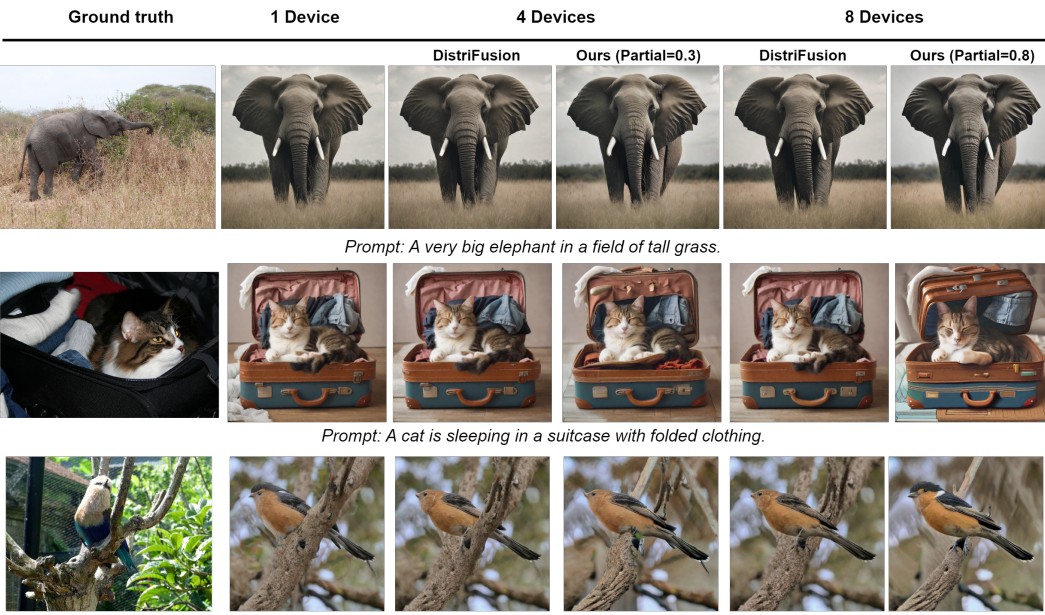

| Ground truth | 1 Device | 4 Devices | | 8 Devices | |
| | | DistriFusion | Ours (Partial=0.3) | DistriFusion | Ours (Partial=0.8) |

*Prompt: A very big elephant in a field of tall grass.*

*Prompt: A cat is sleeping in a suitcase with folded clothing.*

*Prompt: A bird that is on a tree limb.*

Figure 5: Visual comparison of sample images generated by three prompts that work relatively well with PCPP. More examples are available in Appendix B.

### 5.3 GENERATION QUALITY

As shown in Table 2, the images generated using our method PCPP generally achieved lower results than DistriFusion. This difference is an expected trade-off resulting from our communication strategy. By employing point-to-point communication with only neighboring patches, the inference process inherently loses access to a significant amount of contextual information from non-neighboring regions. Nevertheless, PCPP achieved reasonably close PSNR scores compared with DistriFusion, indicating comparable reconstruction quality and noise level. In terms of LPIPS and FID, PCPP performed worse than DistriFusion, meaning that images generated using PCPP deviate more from those generated using one device than DistriFusion.

In addition to numerical evaluations, we visually compared sample images generated by our PCPP method, the original image generated by one device, and the images generated by DistriFusion. As reflected in the metrics, a sizable portion of the generated images by PCPP has distortions and irrational parts. This suggests that having only part or all neighboring patches may not generate a coherent image for all prompts. In Figure 5, we show three prompts that work reasonably well with PCPP. In the elephant example, the images generated by PCPP are visually indistinguishable from those produced by DistriFusion, suggesting that our method can achieve comparable image quality for certain prompts. However, in the cat example, there are noticeable differences between the PCPP and DistriFusion outputs when 8 devices are used, although the PCPP-generated images still maintain high fidelity and coherence. In the bird example, the image generated by PCPP with 8 devices is more similar to that generated by one device in terms of bird color. This shows the randomness in the diffusion process, even with the same random seed. This discrepancy in the quality of generated images suggests that partial attention, which is important for the diffusion process, may not always come from the immediate neighboring parts. More comparisons are available in Appendix B.

### 6 DISCUSSION

In this research, we propose Partially Conditioned Patch Parallelism (PCPP) to accelerate the inference of high-resolution diffusion models leveraging multiple computing devices. Trading generation quality for speed, we achieve better speed-up than DistriFusion - the state-of-the-art implementation of Patch Parallelism (PP) - with less computation and communication. As a result, PCPP decreases

the communication cost by around $70\%$ compared to DistriFusion and achieves $2.36 \sim 8.02\times$ inference speed-up using $4 \sim 8$ GPUs compared to $2.32 \sim 6.71\times$ achieved by DistriFusion depending on the computing device configuration and resolution of generation at the cost of a decrease in image quality. PCPP demonstrates the potential to strike a favorable trade-off, enabling high-quality image generation with substantially reduced latency.

**Limitation and future directions.** The experiments conducted are restricted by the computing resources available. As a result, a maximum of 8 GPUs and a highest resolution of $3840 \times 3840$ are used. The impact of having more GPUs, i.e., more patches, on the quality of generation by PCPP has thus not been studied. We hypothesize that for a larger number of GPUs, the context needed for the generation of a patch may extend beyond immediate neighboring patches, corresponding to a partial value larger than 1. The key insight from this research - reducing collective communication to point-to-point communication with neighboring patches - provides an interesting perspective on the trade-off between inference efficiency and generation quality in diffusion models. During parallel inference, how much and what context does the model really need, and how fresh should those stale activations from the previous denoising step be to achieve similar image quality? These questions deeply about diffusion may be further studied in future research. The inconsistent quality of images generated by partially conditioned attention on parts of neighboring patches suggests that a more involved selection of relevant context is needed as a direction for future research. Furthermore, due to the scope of this research, the effect of PCPP on popular optimization and inference methods for diffusion models, such as ControlNet (Zhang et al., 2023) and LoRA (Hu et al., 2021), is left for future discussion. Previous studies in image generation by diffusion model following a patch-like approach for memory optimization in a single device (Yang et al., 2024; Ding et al., 2024) may also be integrated with PCPP to improve the generation quality while taking advantage of the asynchronous point-to-point communication.

Another potential limitation of this research (and accelerating inference with multiple computing devices in general) is that requiring multiple GPU devices for a single-image generation might seem like an inefficient use of resources. However, we believe this approach has potential use cases in specific areas, such as high-resolution image editing applications that demand high responsiveness. In a traditional single-device setup, generating a high-resolution image could take more than 2 minutes. In contrast, by leveraging PCPP, we can complete the same task in under 15 seconds without any significant decrease in image quality, as shown in Section 5. This significant reduction in latency can be particularly valuable in interactive applications where users expect near-instant feedback, such as real-time image editing tools.

**Ethics statement.** PCPP does not intend to alter the image generated and its meaning. The generated content is primarily determined by the diffusion model and the prompts given, with possible distortions introduced by the parallelizing computing approach. Therefore, the ethical practice of using generation service accelerated by PCPP is ensured by the design of the diffusion model and restrictions on the prompts as input, not as a part of PCPP.

PCPP can be applied to accelerate the generation of images by diffusion models leveraging multiple resources. Compared to the previous state of the art approach, PCPP reduces computation and communication for the same inference, thus contributing to energy saving for image generation. The speed-up might be helpful for many applications requiring a timely response given a prompt. In addition, PCPP might also be applied to diffusion models that generate data structures other than images. We wish PCPP can be a useful tool for researchers, designers, and users of diffusion models.

**Reproducibility statement.** We provided a detailed description of the method and its implementation in Section 3. In addition, more implementation details are described in Appendix A. In Section 4, we included the computing device, dataset, and evaluation metrics for reproducing the experiments. We also plan to release the code and scripts necessary to reproduce our results.

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

## A    PARALLEL IMPLEMENTATION

**Design Choice on Synchronization.**    In our current implementation, for any device $i$, we explicitly wait for the asynchronous operations of layer $l$ at step $t$ to complete at the beginning of layer $l+1$ at the same step $t$, even if we won't be using them until the layer $l$ at the next denoising step $t-1$. One might wonder why we don't call the .wait() method when the activations are actually needed. We have considered this initially. However, the NCCL (NVIDIA Collective Communications Library) backend does not support tags in the isend and irecv operations. This means that if we allow isend and irecv operations of multiple layers to be in-flight at the same time, the receiving device won't be able to identify which tensors are from which layers. As such, we decided to call the .wait() operation a bit earlier, so that it guarantees that at any given time, only one group of asynchronous send and receive operations between any two devices are in-flight. This ensures that the receiving device can properly match the incoming tensors to the correct layers.

**Avoiding Potential Deadlocks.**    Since one device can simultaneously fire up to two pairs of isend and irecv calls to its two neighbors, if we were to issue these operations individually, there are risks of deadlocks when the order of isend and irecv does not match each other. To address this issue, we decided to use torch.distributed.batch_isend_irecv, where we can issue all the send and receive operations for a given step in a single call, and the library handles the coordination and matching of the tensors on the receiving side. This not only avoids the potential for errors or mismatches, but also simplifies the bookkeeping and management of the asynchronous requests in our code. Instead of maintaining a separate list of isend and irecv requests for each layer, we can work with a single batch_isend_irecv request object, which makes the logic more concise.

**Batched Communication of Layer Activations.**    One potential future work is to explore accumulating the activations for a few continuous layers before initiating the communication. In our current implementation, we send the activations for each layer individually using dist.batch_isend_irecv as soon as they are ready, as described in Section 3.2. However,

we could potentially optimize this by accumulating the activations for a couple of layers before triggering the communication. This would allow us to send a batch of activations for multiple layers at once, rather than sending them one layer at a time. This would reduce the overall communication overhead by amortizing the costs of setting up and tearing down the communication across multiple layers. Additionally, sending larger batches of activations may allow the underlying NCCL communication libraries to optimize the transfers more effectively.

## B  GENERATION RESULTS

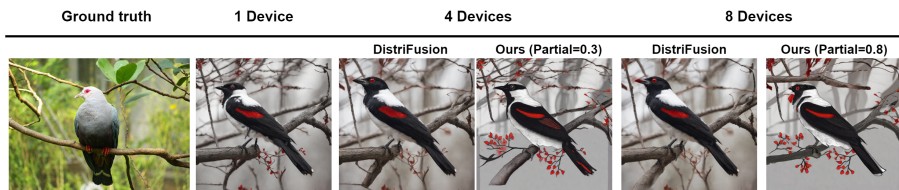

Figure 6: *Prompt: a black and white bird with red eyes sitting on a tree branch.* The style of the generated image using PCPP changes from a realistic image to a painting.

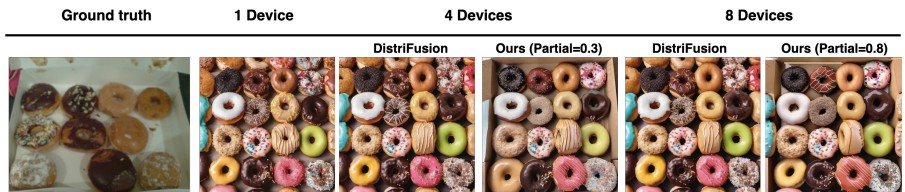

Figure 7: *Prompt: A box of donuts of different colors and varieties.* The image generated using PCPP is almost identical to the image generated using DistriFusion.

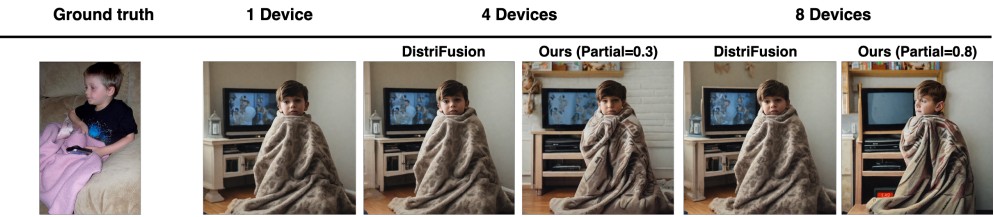

Figure 8: *Prompt: A boy covered up with his blanket holding the television remote.* The image generated using PCPP slightly deviates from the original in terms of the TV background and the direction the boy is facing.

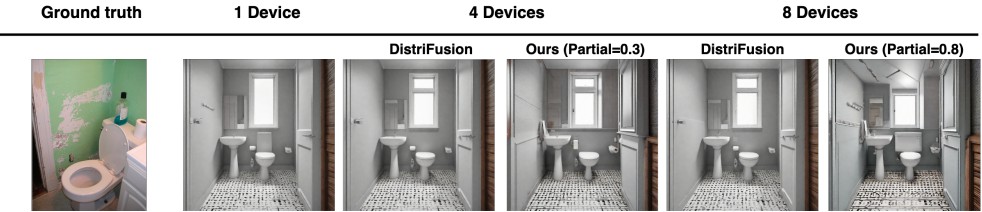

Figure 9: *Prompt: The toilet is near the door in the bathroom.* The image generated using PCPP has a slightly distorted rendering of the bathroom ceiling and floor.

