# OpenReview forum: "Partially Conditioned Patch Parallelism for Accelerated Diffusion Model Inference"
_ICLR.cc/2025/Conference — ICLR 2025 Conference Withdrawn Submission_

### Official Review · Reviewer_9qZs · 2024-10-18

**Soundness:** 3
**Presentation:** 2
**Contribution:** 1
**Rating:** 3
**Confidence:** 3

**Summary:**

*Problem*:
Patch parallelism employs multiple GPUs to generate an image. However, PP can be improved in terms of efficiency.

*Core ideas*:
1. Let's communicate only part of the neighbor patch (Figure 1).
2. Let's communicate asyncronously; t=10 and t=9 are largely similar (Figure 2).
3. Let's calculate attention with only neighbors (Figure 3).

*Experiment*:
Run SDXL on 8 A100s, making a 4k*4k image in a very short time.

**Strengths:**

1. *Addresses a practical problem*: The use of multi-GPU systems is important, and the knowledge presented in this paper could be applied to other fields as well. Since video generation requires significant runtime, the techniques proposed in this paper could potentially be extended for use in video generation.

2. *Significantly reduces communication cost*: In systems where communication cost is a bottleneck, the proposed method can be highly beneficial.

**Weaknesses:**

**Limited Novelty**: This work is built on top of DistriFusion. Some core ideas are quite trivial; for example, arXiv:2401.05735 used somewhat similar ideas to Figures 2 and 3.

**Marginal gain**: Although the communication cost is reduced significantly (Table 1), the latency is not so different from DistriFusion (maybe because of the Amdahl's law). Moreover, the quality of the image is degraded (Table 2).

**Small scope**: This work only deals with SDXL, and many novelties actually have come from the details of SDXL. For example, the default sampler of SD2.1 is DPM-Solver++ (Lu et al), with much fewer steps than 50. This work assumes that $A^{(i)}_{l,t}$ is not so different from $A^{(i)}_{l,t-1}$ or $A^{(i)}_{l,t+1}$, which is not true in very recent samplers. (DPM-Solvers, and many few-step sampling methods in LCM or RF-based models)

**Questions:**

1. Why did you use cfg 5? The default setting in SDXL is 8.
2. What did you cite in L094? It should have been the classifier-free guidance; you have cited DPM-Solver++.

---

### Official Review · Reviewer_97Pd · 2024-10-29

**Soundness:** 3
**Presentation:** 2
**Contribution:** 2
**Rating:** 3
**Confidence:** 4

**Summary:**

This paper proposed Partially Conditioned Patch Parallelism (PCPP)  using GPUs to reduce computation by conditioning on parts of the neighboring patches in each diffusion steps. The method is built to improve existing DistriFusion.

**Strengths:**

This paper proposed to  reduce computation in inference by conditioning only on parts of the neighboring patches in each diffusion steps.

**Weaknesses:**

I believe the contribution in this paper is not significant enough to be published in this conference. There is no sufficient novelty in the idea presented in this paper.
The paper is based on the existing DistriFusion and what they did is to identify areas that can be further parallelized.

The base for them to propose their improvement is similar to
They authors claimed their method is "using  the fact that the difference between the images in adjacent diffusion steps is nearly zero".

This is similar to what was stated in DistriFusion:
"... we observe the high similarity between the input from adjacent diffusion steps"
As a result, the most significant improvement introduced is to use the parts of the neighboring patches in each diffusion steps. As it shows in the paper, it improved speed in someway, and it also incurred some problems.

**Questions:**

1. Figure 1. The caption describing figure 1 is confusing.
2. Page 4, eqn. (1). The expression is vague while the authors referred to upper or lower ph
3. There is no discussion on the selection of s in eqn. (2) in page 4, around line 191.

4. The authors claimed that "The total communication cost is also greatly reduced due to the replacement of collective communication with point-to-point communication" however, there is no explanation on why

5. The explanation in Section 3.2 and 3.3 should be improved.

---

### Official Review · Reviewer_szu9 · 2024-11-03

**Soundness:** 2
**Presentation:** 2
**Contribution:** 2
**Rating:** 3
**Confidence:** 3

**Summary:**

This paper introduces **Partially Conditioned Patch Parallelism (PCPP)**, a method to accelerate sample generation in diffusion model inference. The core idea is to divide an image into non-overlapping patches and use multiple devices to generate each patch independently. Each patch’s generation is conditioned only on its neighboring patches, reducing inter-device communication compared to conditioning on the entire image.


The authors compare the proposed PCPP method with DistriFusion, showing a 70% reduction in inter-device communication. Inference speed for DistriFusion ranges from 2.32 to 6.71 times faster than the original generation speed (varying by device count and image resolution), while PCPP achieves a speedup of 2.36 to 8.02 times.

**Strengths:**

+ The challenges with the image generation using diffusion model (speed) is relevant to the community and the idea explored by the paper is interesting.

**Weaknesses:**

### Writing and Readability

- **Writing Clarity**: The paper does not read easily and it could certainly be improved.
    - **Abstract, line 12**: "However, the inference speed of diffusion models is limited by the slow sampling process, restricting its use cases."
    - **Line 38**: "Doing a diffusion process in the latent space, in which dimensionality can be much smaller than the image space, requires significantly lesser computation to train a diffusion model and make inferences with it than in the image space."

- **Suggestion**: The authors can improve the writing of Section 2.1 greatly.

### Literature Review

- **Missing Literature**: A comparison with relevant literature is lacking. For instance, *PipeFusion* proposed a similar approach for image generation using diffusion transformer models, also employing patch parallelism.

- **Reference**: Please include the PipFusion in the literature review as well.


Jiannan Wang, Jiarui Fang, Aoyu Li, PengCheng Yang, "PipeFusion: Displaced Patch Pipeline Parallelism for Inference of Diffusion Transformer Models", arXiv preprint arXiv:2405.14430.

### Experimental results

- **Experimental Choices**: While the idea is interesting and valuable for diffusion model applications in imaging, the experiments could be more thorough. For instance:
    - **Partial Choice**: The rationale for choosing 0.3 as the partial is unclear. It would be beneficial to explain why this value was selected over others.
    - **Impact of Different Partials**: An analysis of varying partials would help readers understand the trade-offs in quality, latency, and inference speed. Including a comprehensive table showing these trade-offs for different partial values would aid in making more informed decisions regarding the PCPP method.

### Results and Observations

- **Results Feasibility**: The trade-offs between inference time and quality do not appear compelling enough to motivate adoption of PCPP. Specifically:
    - **FID and PSNR**: The FID is relatively high compared to DistriFusion, and PSNR is lower. Inference speed improvements across resolutions do not consistently offer an advantage.

**Questions:**

1. **Details of Table 2**: Could the authors clarify the number of devices and specifics of the partial value used in Table 2?

2. **PSNR Metric for Figure 5**: Could the authors report the PSNR metric for the images in Figure 5? The reported value of 28.8 seems high for the displayed images, particularly if they represent favorable samples with PCPP.

---

### Note · Authors · 2024-12-05

**Comment:**

We thank all the reviewers for their time and effort in giving meaningful and constructive feedback. We will continue iterating on our research to provide better value to the community.

**Withdrawal Confirmation:**

I have read and agree with the venue's withdrawal policy on behalf of myself and my co-authors.